# A Comparative Kidney Transcriptome Analysis of Bicarbonate-Loaded *insrr*-Null Mice

E. A. Gantsova [1,2], O. V. Serova [1], D. Eladari [3], D. M. Bobrovskiy [4], A. G. Petrenko [1], A. V. Elchaninov [2] and I. E. Deyev [1,*]

1   Shemyakin-Ovchinnikov Institute of Bioorganic Chemistry of the RAS, Moscow 117997, Russia; gantsova@mail.ru (E.A.G.); oxana.serova@gmail.com (O.V.S.); petrenkoag@gmail.com (A.G.P.)
2   Avtsyn Research Institute of Human Morphology of Federal State Budgetary Scientific Institution "Petrovsky National Research Centre of Surgery", Moscow 117418, Russia; elchandrey@yandex.ru
3   Service de Médecine de Précision des Maladies Métaboliques et Rénales, CHU Amiens-Picardie, Université de Picardie Jules Verne, 80054 Amiens, France
4   School of Life Sciences, École Polytechnique Fédérale de Lausanne (EPFL), 1015 Lausanne, Switzerland
*   Correspondence: deyevie@ibch.ru; Tel.: +7-(495)-335-41-77

**Abstract:** The maintenance of plasma pH is critical for life in all organisms. The kidney plays a critical role in acid–base regulation in vertebrates by controlling the plasma concentration of bicarbonate. The receptor tyrosine kinase IRR (insulin receptor-related receptor) is expressed in renal β-intercalated cells and is involved in alkali sensing due to its ability to autophosphorylate under alkalization of extracellular medium (pH > 7.9). In mice with a knockout of the *insrr* gene, which encodes for IRR, urinary bicarbonate secretion in response to alkali loading is impaired. The specific regulatory mechanisms in the kidney that are under the control of IRR remain unknown. To address this issue, we analyzed and compared the kidney transcriptomes of wild-type and *insrr* knockout mice under basal or bicarbonate-loaded conditions. Transcriptomic analyses revealed a differential regulation of a number of genes in the kidney. Using TaqMan real-time PCR, we confirmed different expressions of the *slc26a4*, *rps7*, *slc5a2*, *aqp6*, *plcd1*, *gapdh*, *rny3*, *kcnk5*, *slc6a6* and *atp6v1g3* genes in IRR knockout mice. Also, we found that the expression of the *kcnk5* gene is increased in wild-type mice after bicarbonate loading but not in knockout mice. Gene set enrichment analysis between the IRR knockout and wild-type samples identified that *insrr* knockout causes alterations in expression of genes related mostly to the ATP metabolic and electron transport chain processes.

**Keywords:** IRR; receptor tyrosine kinases; alkaline pH; kidney; intercalated cells; alkalosis; acid–base balance

## 1. Introduction

Blood pH is kept constant within a narrow range, as even a small change in pH could greatly affect cells, tissues, and the whole organism. The normal arterial blood pH value is $7.40 \pm 0.04$, and a pH < 7.36 defines acidosis, whereas a pH > 7.44 defines alkalosis. The $CO_2/HCO_3^-$ buffer system plays a critical role in the stability of extracellular pH even though the dissociation constant of this system is quite far below the usual extracellular pH value, which should make it a relatively ineffective buffer system. However, the two components of this buffer system are controlled independently via two remarkably effective homeostatic mechanisms, (1) the respiratory (lung) and (2) urinary (kidney) systems, which regulate plasma $pCO_2$ and $[HCO_3^-]$, respectively. The kidneys regulate systemic acid–base balance by controlling the reabsorption of $HCO_3^-$ filtered at the glomerulus and by excreting the daily acid load generated metabolically. About 70% of filtered $HCO_3^-$ is reabsorbed by proximal tubule (PT) cells, 25% in the thick ascending limb, and the remaining 5% in the collecting duct. Collecting ducts perform either reabsorption or secretion of bicarbonate depending on the needs of the organism.

The regulation of acid–base homeostasis involves a broad variety of molecular mechanisms including membrane transporters and signaling pathways. The presence of endogenous pH-sensors, i.e., molecules that detect changes in pH and which in turn adapt physiological functions have been discovered recently [1]. One of those pH sensing molecules is the insulin receptor-related receptor (IRR). IRR is a receptor tyrosine kinase (RTK) that belongs to the insulin receptor family which also includes insulin receptor (IR) and insulin-like growth factor receptor (IGF-R). Ligands for IR and IGF-IR are insulin and insulin-like growth factors I, II (IGF-I, II), respectively. Attempts to find a natural ligand for IRR of protein or peptide nature has remained unsuccessful. However, in 2011, we reported that IRR is activated by alkaline extracellular pH. The activation of IRR by alkali is specific, reversible, and dose dependent [2], and leads to insulin receptor substrate 1 (IRS-1) phosphorylation and cytoskeleton remodeling [3].

Unlike IR and IGF-IR receptors, which are expressed in a wide range of tissues and cells, the pattern of IRR expression is highly restricted. The largest amount of IRR was found renal β-intercalated cells (β-IC) of cortical collecting ducts. β-ICs are renal epithelial cells involved in bicarbonate secretion into urine, a process that involves the apical $Cl^-$/$HCO_3^-$ exchanger pendrin and a basolateral V-ATPase [4]. Using mice with the targeted inactivation of *insrr* gene, we demonstrated that IRR controls renal base excretion. We found that knockout of the *insrr* gene leads to a decreased expression of the bicarbonate exchanger pendrin. Furthermore, *insrr* knockout mice were unable to cope with exogenous alkali load and developed metabolic alkalosis [2,5]. In the latter study, we did not assess whether IRR might also be involved in other renal physiological mechanisms or functions. Therefore, we performed the analyses of large-scale sequencing of the kidney transcriptome of wild-type and *insrr* knockout mice under basal or alkali-loaded conditions to determine which biological processes might be affected by IRR in the kidneys.

## 2. Materials and Methods

Mouse strains and genotyping *Insrr*−/− knockout mice were generated and described previously [2,6]. A strain of knockout mice for the *insrr* gene was obtained earlier and studied in the works of T. Kitamura and S. Nef [6,7]. In all the experiments described in this work, we used only littermate 3–4 months old female mice [8]. The genotype was verified by PCR with genomic DNA of mice to determine the presence of a knockout or wild-type allele. The genotype of 191 mice was determined. Of these, there were 47 homozygous wild-type mice, 55 homozygous knockout mice, and 89 heterozygotes. Wild-type and *insrr* knockout mice (4 animals in each group) were either fed water ("Normal conditions") or alkali loaded with 0.28 M $NaHCO_3$ administrated in the drinking water for 7 days. This treatment has been shown to induce mild metabolic alkalosis in mice or rats [9,10].

To maintain a pure genetic background and minimize the effect of accumulation of mutations, the method of littermate breeding was used. *Insrr*−/− knockout mice and their control littermates *insrr*+/+ were obtained by crossing *insrr*+/− heterozygotes. The genotype of the mice was determined by PCR. Genomic DNA used as a template was isolated from mouse tails using a method described in [11]. The wild type allele was detected using primers:

mwt1 (5′-GCAAGCTACACAGGCTCGAGGG-3′)
mwt2 (5′-TGGGTTCTGATCCTCTCAAGGAG-3′).
Primers were used to identify the knockout allele.
ko1 (5′-CAAAACCAAATTAAGGGCCAGCTC-3′)
ko2 (5′-AGCCTGAAGACCCTCGTCGACT-3′)

Animal experiments, bicarbonate loading. All experiments with animals were performed according to the protocol of the Institutional Animal Care and Use Committee (IACUC), approved by the Bioethics Commission of the Shemyakin–Ovchinnikov Institute of Bioorganic Chemistry of the Russian Academy of Sciences (IBCH RAS) (project identification code No. 341), and handled in accordance with the Animals (Scientific Procedures) Act 1986 and Helsinki Declaration. Laboratory animals were kept with free access to food

and water, at constant room temperature (24 °C $\pm$ 1 °C) with 12 h light/dark cycle. All experiments were performed on 3–4-month-old female *insrr+/+* and *insrr−/−* mice. Animals were randomly assigned to 4 groups of 4 mice: (1) *insrr+/+* mice under normal conditions (WT_water), (2) alkali-loaded *insrr+/+* mice (WT_bicarb), (3) *insrr−/−* mice under normal conditions (KO_water), (4) alkali-loaded *insrr−/−*mice (KO_bicarb). The mice were given deionized water under normal conditions. To induce a mild metabolic alkalosis the mice received drinking water supplemented with 0.28 M sodium bicarbonate (NaHCO$_3$) for 7 days. Before kidney sampling mice were fasted for 12 h with free access to water (or NaHCO$_3$ solution).

RNA sequencing of mouse kidney. The mice were anesthetized with combination of zoletil (20 mg/kg body weight) and xylazine (5 mg/kg body weight) in 0.9% NaCl solution. The mice were then euthanized via cervical dislocation. The kidneys were removed from the abdominal cavity and washed in sterile PBS solution. For RNA sequencing, a segment of the kidney with a thickness of 3 mm was cut from the middle section along the horizontal plane and placed in RNAlater buffer. All RNA purifications and RNA-seq analyses were performed by Genoanalytica (Moscow, Russia). Briefly, total RNA was extracted from the samples with TRIzol Reagent solution (Invitrogen, Carlsbad, CA, USA) according to the manufacturer's protocol. Quality of RNA was checked with BioAnalyser and RNA 6000 Nano Kit (Agilent, Santa Clara, CA, USA). PolyA RNA was purified with Dynabeads® mRNA Purification Kit (Ambion, CA, USA). The Illumina library was made from polyA NEBNext® Ultra™ II RNA Library Prep (NEB, Ipswich, MA, USA) according to the manual. Sequencing was performed on HiSeq1500 with 50 bp read length. At least 20 million of the reads were generated for each sample.

Reads were mapped onto the genome using Star Aligner and the fold changes were calculated using DEseq2.0 software. Raw RNA-seq data were deposit to European Nucleotide Archive under GSE200638 number (Available online: https://www.ncbi.nlm.nih.gov/geo/query/acc.cgi?acc=GSE200638 (accessed on 13 April 2022)).

RNA-Seq reads were mapped to the reference mouse genome (mm10) using STAR [12]. Uniquely mapped reads (Ensembl v. 92) were used to quantify the expression levels. Differential expression analysis was performed using edgeR [13]. Principal component analysis (PCA) was performed on the CPM (counts per million) matrix exported from edgeR. For the functional enrichment analysis and weighted gene co-expression network analysis (WGCNA) [14], the RNA-seq reads were quality controlled and quantified using salmon selective alignment with unstranded library types and parameters, validateMappings, and –seqBias, using the entire Mus musculus reference genome (GRCm39, GENCODE M27) as the decoy sequence [15,16]. The resulting abundances were also processed using edgeR, with the offsets being calculated with tximport [17]. Only protein-coding genes, according to BioMart [18], were considered for analysis. The transcripts expressed in less than 25% of samples or having less than 10 counts in each sample or total count less than 100 were filtered out. After all these filtration steps, our dataset contained 13,430 genes. Genewise differential expression (DE) analysis was performed using an edgeR quasi-likelihood $F$-test (QL $F$-test) under a negative binomial generalized log-linear model. The test was performed either between knockout and control samples (separately for different treatments), or between the samples from bicarbonate and water treated mice (separately for knockout and wild-type mice). A functional enrichment analysis was performed using fgsea [19] based on the log-fold change (logFC) using clusterProfiler R package [20]. Finally, WGCNA was performed on a reads per kilobase per million (RPKM) matrix exported from edgeR. To build a signed network when performing the WGCNA we used biweight midcorrelation for co-expression similarity and soft thresholding power of 14. Hierarchical clustering with adaptive-height tree cut was performed on a topological overlap matrix with the parameters deepSplit = 1 and minClusterSize = 20, in which module eigengenes were calculated. To assess how gene modules correspond to sample groups, we used linear regression with the following independent variables, genotype (KO or WT), treatment (bicarbonate or water), and their interaction. To annotate the modules,

we selected genes from the modules which turned out to be linked to IRR knockout, and performed Gene Ontology clusterProfiler overrepresentation analysis for those which had a Pearson correlation coefficient with module eigengene greater than 0.8 and QL *F*-test FDR between KO and WT less than 0.05 under either bicarbonate or water treatment.

RNA extraction and cDNA obtaining. Kidneys were removed as described previously, washed in sterile PBS solution, cut horizontally into two parts, and placed in TRIzol reagent solution (Invitrogen). The tissue was then homogenized with Lysing matrix D ceramic beads (MP Biomedicals) using a FastPrep-24 Classis bead beating grinder and lysis system (MP Biomedicals). The total RNA from mouse kidneys was isolated using TRIzol reagent (Invitrogen, USA) according to the manufacturer's protocol. To remove contaminants of genomic DNA, total RNA samples were treated with DNase I (Thermo Scientific, Waltham, MA, USA) according to the manufacturer's protocol. The quality of the isolated RNA samples was verified using 2% agarose gel electrophoresis in TBE buffer (Tris-borate-EDTA) by the presence of clear 28S and 18S RNA bands. cDNA was obtained by a reverse transcription reaction using a RevertAid RT Reverse Transcription Kit (Thermo Scientific, Massachusetts, USA) and random hexamer primers according to the manufacturer's protocol.

TaqMan Real-time PCR. Real-time quantitative PCR was carried out using mouse kidney cDNA samples as a template. Primers and a specifically annealing probe for each gene were selected using GeneRunner, OligoExplorer, and Oligo Analyzer tools. The Evrogen and Lumiprobe companies (Russia) synthesized primers and probes. The probes contained a FAM fluorescence label (Fluorescein amidites) at the 5'-end, and a BHQ1 quencher (Black Hole Quencher 1) at the 3'-end. The ap3d1 gene was used as a reference gene according to the literature data [21]. For each pair of primers, the reaction efficiency (E) was calculated using the formula, $E = 10 - 1/k$, where k is taken from the straight-line equation, $CT = k \log P0 + b$, in which P0 is the concentration of cDNA; CT is the number of cycles in which the fluorescence reaches the threshold level, and the values of k and b are obtained from a linear approximation of the experimental data. The efficiency of the reaction for the target and reference genes was 2, which corresponds to the maximum efficiency of the reaction. A TaqMan real-time PCR with each cDNA sample was performed in three replicas using a DTprime instrument (DNA-Technology, Russia). The reaction parameters were as follows: preliminary denaturation at 95 °C–5 min; 45 amplification cycles: 95 °C–15 s, 60 °C–10 s, 72 °C–10 s. A normalized expression (NE) was calculated as described in [22]. The data obtained for animals of two genotypes corresponded to the criterion of normal sampling (the Shapiro–Wilk test). Real-time PCR data were statistically processed applying GraphPad Prism version 8.0.1 software (GraphPad Software, San Diego, CA, USA) using Student's *t*-test. Differences were considered statistically significant at $p < 0.05$.

Intraperitoneal Glucose Tolerance Test. Wild-type and *insrr* knockout mice under normal conditions were used in experiment (7 mice per group). All procedures were carried out as describe in protocol [23]. A solution of glucose (2 g/kg of body weight) was administered by intraperitoneal injection to mice fasted overnight. Then blood was taken from the tail vein and plasma glucose concentration was measured at various time points over the next 2 h (zero-point, 15 min, 30 min, 60 min and 120 min after injection). The trapezoidal rule was used to determine the area under the curve (AUC). AUC was calculated for each mouse. Next, we determined the mean value ± SEM for each group. The statistical significance of the difference between the means of the two populations was determined using a two-tailed Student's *t*-test. A *p*-value < 0.05 was considered statistically significant.

## 3. Results

Comparative analysis of the kidney transcriptomes of wild-type (WT) and *insrr* knock-out (KO) mice using RNAseq was performed. Since we have previously shown that IRR is important for the renal adaptation to alkali loading, we analyzed and compared

the transcriptomes under normal conditions and under alkali-loading conditions. Thus, four groups of data were obtained, WT under normal conditions, WT under alkali-loading conditions, KO under normal conditions and bicarbonate loading. Each group included four mice. Also, to avoid additional acid load due to the feeding process of the mice, which can lead to additional variation in the acid–base status of mice, the mice were fasted for 12 h with free access to water (or NaHCO$_3$ solution).

Under normal conditions, KO mice displayed 2316 differentially expressed genes (DEGs) compared to the wild type, including 1348 upregulated and 968 downregulated DEGs (Figure 1A). After bicarbonate loading, expression of 2879 genes were significantly changed in *insrr* knockout mice compared to the wild type, with 1504 upregulated and 1375 downregulated DEGs (Supplementary Data File S1). Comparing mice from same genotype after bicarbonate loading or under physiological conditions, we found 136 DEGs in wild-type mice (39 upregulated and 97 downregulated) and 157 DEGs (35 upregulated and 122 downregulated) in *insrr* KO mice (Figure 1A). The overall structure of the RNA-seq dataset was assessed using principal component analysis (PCA) (Figure 1B). As shown by the PCA component plot, bulk RNA-seq data of wild type and *insrr* knockout groups of mice clearly differed from each other.

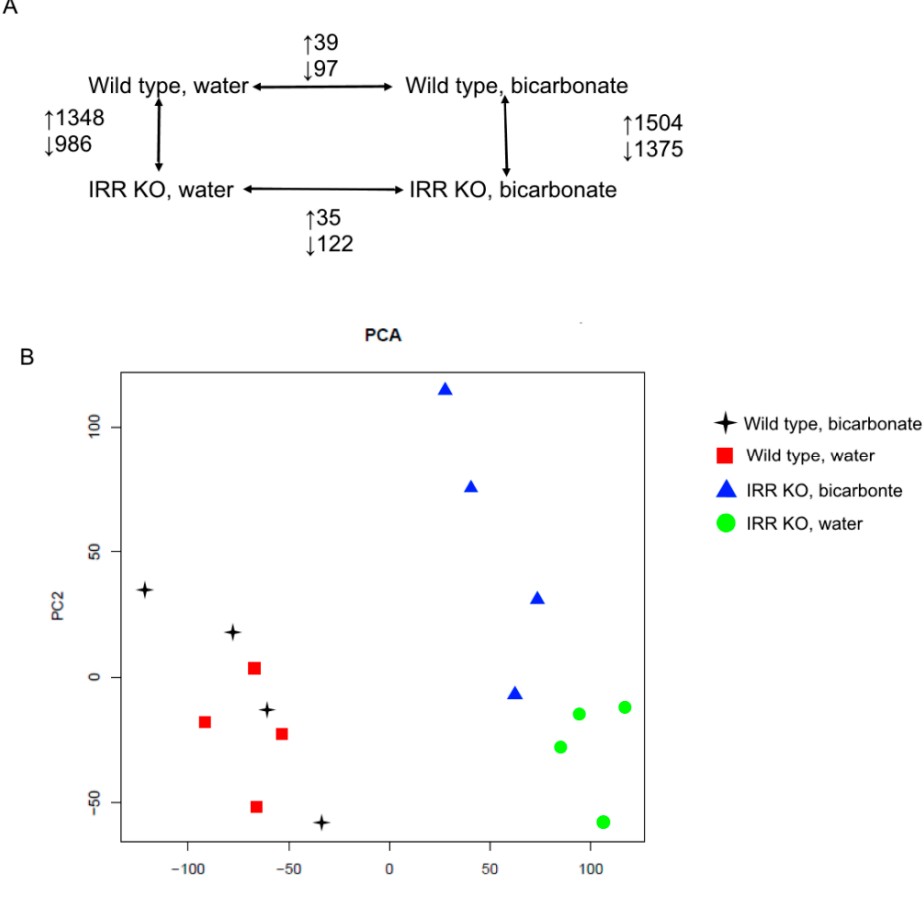

**Figure 1.** Number of differentially expressed genes (**A**), PCA of the samples based on the RPKM matrix, two first principal components (**B**).

Gene Ontology fast gene set enrichment analysis (GO fgsea) based on the log-fold change (logFC) between the *insrr* knockout and wild-type samples under normal condition identified that knockout causes alterations in expression of genes related to the electron transport chain and several other processes (Figure 2), like ATP synthesis, protein synthesis and electron transport. To be precise, the genes involved in oxidative phosphorylation and ribosomal proteins are substantially upregulated after *insrr* knockout, whilst the genes

related to several transmembrane signaling pathways seem to be downregulated. Gene Ontology fgsea of the differentially expressed genes between the knockout and wild-type groups under bicarbonate treatment shows generally the same patterns (Figure S1).

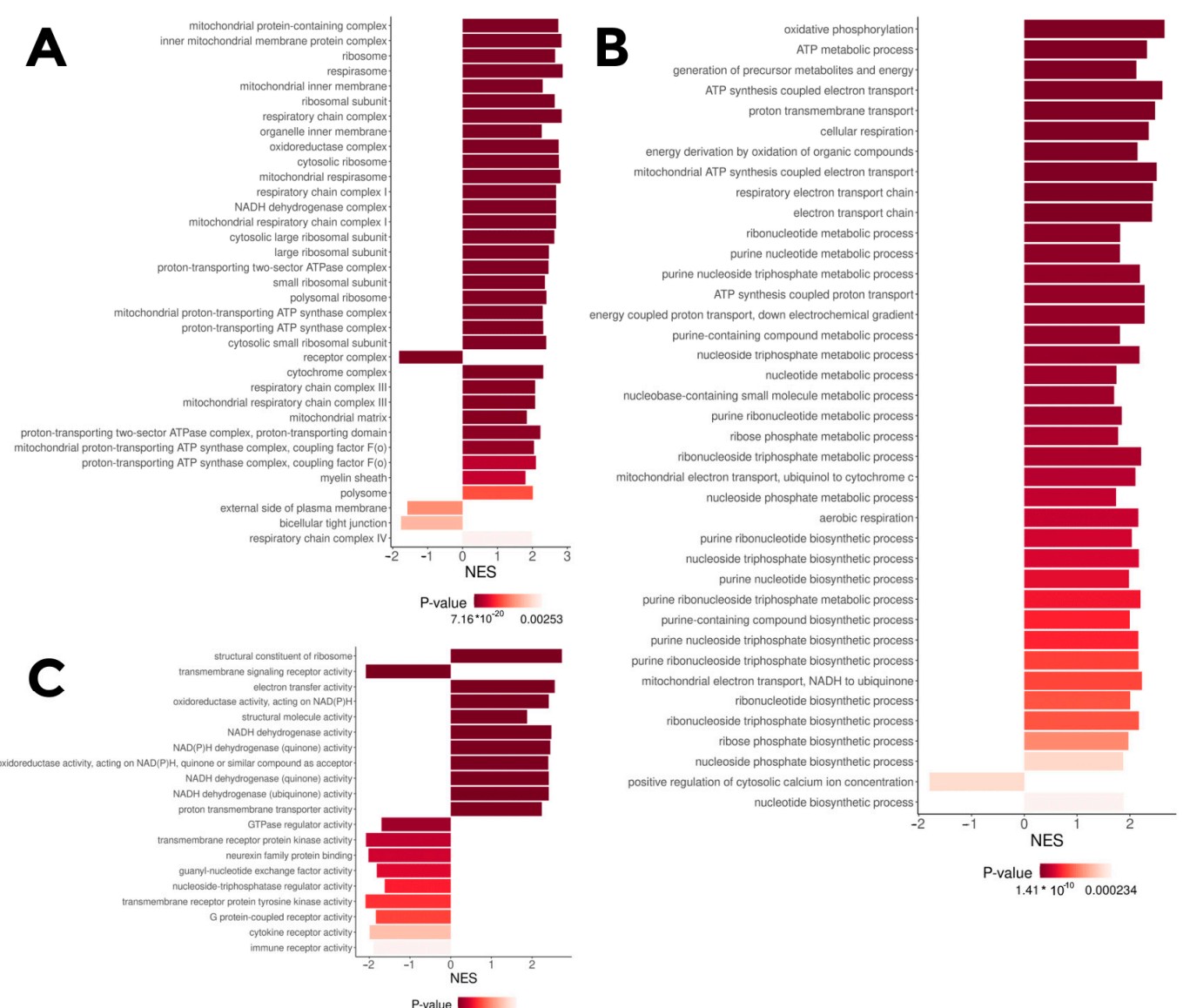

**Figure 2.** GO terms for which the IRR knockout differential genes under water-treatment are enriched (fgsea on logFC, adjusted $p < 0.05$), NES—normalized enrichment scores (positive—upregulated in knockout, negative—downregulated). (**A**)—GO cellular components; (**B**)—GO biological process; (**C**)—GO molecular function.

Using weighted gene co-expression network analysis (WGCNA) we identified seven gene modules, composed of genes with a correlated expression across samples (Figure 3A). For the eigengenes, which summarize the expression patterns of each of the modules, we fitted a linear model with three independent variables, genotype, treatment, and their interaction. Figure 3C shows the correspondence between the eigengene values and the sample groups. The three largest modules (turquoise, blue, and brown) correlated significantly with *insrr* knockout ($p < 0.05$, linear regression with Bonferroni correction for the number of modules). The brown module also showed relatively significant dependence on the interaction of the *insrr* knockout and bicarbonate treatment ($p = 0.03$ without correction). To annotate the modules, from the turquoise, blue, and brown module genes, we selected DEGs (QL $F$-test FDR < 5%, either under bicarbonate or water treatment), which were

most correlated with the module eigengenes (Pearson correlation coefficient > 0.8). The resulting gene lists were used for GO overrepresentation analysis. Consistent with simple differential expression analysis, the turquoise module, corresponding to genes upregulated in *insrr* knockout mice compared to the wild-type animals, was strongly linked to the oxidative phosphorylation (Figure S2). However, the blue module representing genes downregulated in *insrr* knockout mice, related to many other biological processes, such as ribosome biogenesis, non-coding RNA processing, mitochondrial structure and protein folding (Figure S3). In the case of the brown module, over-representation analysis showed no consistent enrichment.

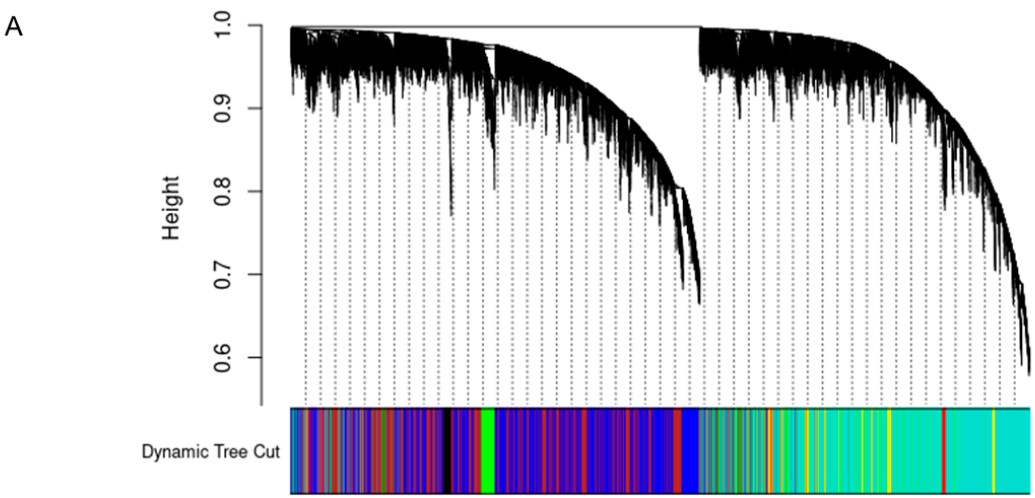

| Module | turquoise | blue | brown | yellow | green | red | black |
|---|---|---|---|---|---|---|---|
| **N of genes** | 4726 | 3774 | 3136 | 916 | 434 | 261 | 183 |

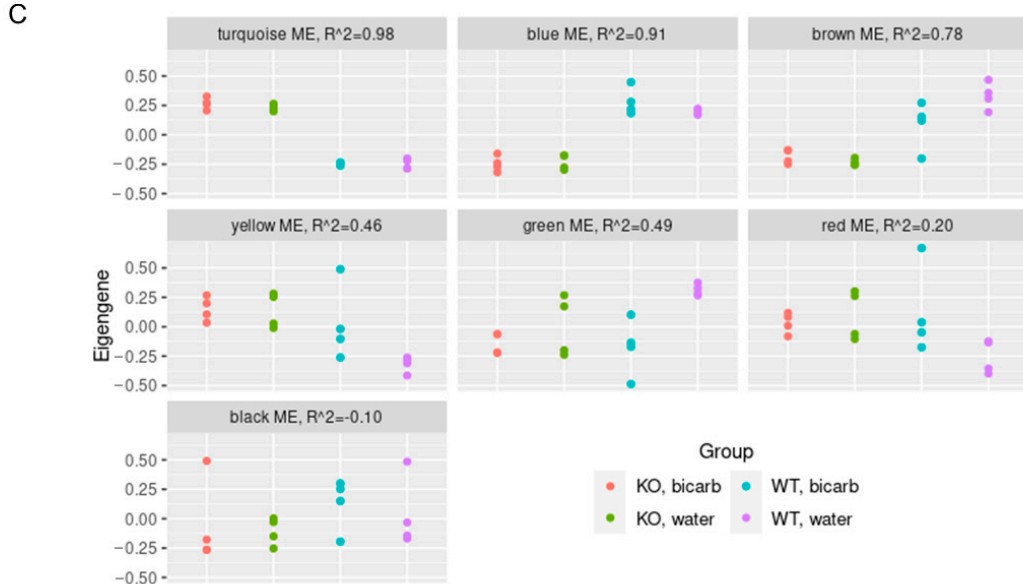

**Figure 3.** WGCNA gene dendrogram and module colors (**A**), size of the WGCNA gene module (**B**), WGCNA module eigengenes values by treatment groups, with R2 of the linear model with the following independent variables: genotype (KO or WT), treatment (bicarbonate or water) (**C**).

To find genes which expression depends on the presence of only IRR in the kidneys we built the point diagram of DEGs log2(Fold change KO water/WT water) versus log2(Fold change KO bicarbonate loading/WT bicarbonate loading) for each of DEGs which is found between WT and KO mice in basal state and after bicarbonate loading (Figure 4A). This analysis revealed that mostly DEGs expression significantly depends on only IRR expression (Supplementary Data File S2). Supplementary Table S1 shows top 25 DEGs which changes expression in mouse kidney after *insrr* knockout, including *gapdh* gene.

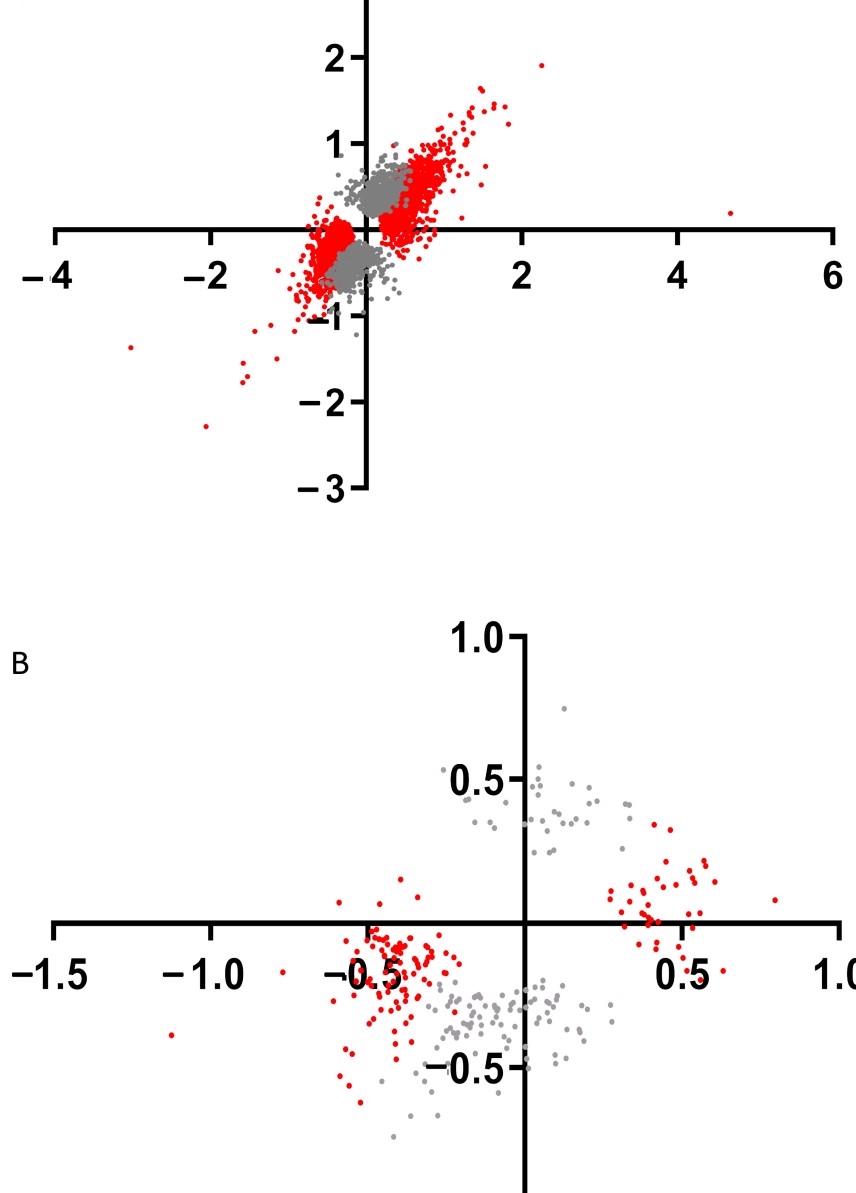

**Figure 4.** (**A**) Graphical plot log2(Fold change KO water/WT water) versus log2(FoldChange KO bicarbonate loading/WT bicarbonate loading) for each of DEGs found between WT and KO mice in basal state and after bicarbonate loading. Red color indicates only DEGs between WT and KO mice under basal state. (**B**) Graphical plot log2(Fold change WT bicarbonate loading/WT water) versus log2(FoldChange KO bicarbonate loading/KO water) for each of DEGs found after bicarbonate loading in WT and KO mice. Red color indicates only DEGs in WT after bicarbonate loading. Grey color indicates remaining DEGs.

Using this approach, we tried to find genes whose expression depends on the presence of IRR in the kidneys and were significantly changed via bicarbonate loading in wild-type mice. We made a point diagram of DEGs log2(Fold change WT bicarbonate loading/WT water) versus log2(Fold change KO bicarbonate loading/KO water) for each of DEGs found in WT and KO mice after bicarbonate loading (Figure 4B). This analysis revealed several dozen of DEGs whose expression significantly depends on IRR expression after bicarbonate loading (Supplementary Data File S2) and the top 25 of these DEGs are shown in Supplementary Table S2. Among these top genes are *Kcnk5*, *Atp5k* and *Atp6v1g3* genes.

To confirm these data, we checked expression of some DEGs using TaqMan real-time PCR. cDNA samples from additional wild type and KO mice. The same four groups of animals were also used in these experiments, WT physiological conditions, WT bicarbonate loading, KO physiological conditions, KO bicarbonate loading. The classic housekeeping *gapdh* gene, as well as another common housekeeping gene, *rps7*, did not fit as reference genes. According to NGS data, they strongly change their expression in *insrr* knockout mice. Decreased *gapdh* gene expression in mouse kidneys also was confirmed via TaqMan real-time PCR and Western blotting described in our previous work [24]. Using data from [21] for reference gene search we selected *ap3d1* and *csnk2a2*. Their expression did not change in animals of two genotypes under different conditions, according to NGS data. The *ap3d1* gene showed a higher level of expression and was chosen as the main reference gene for calculating real-time PCR data. For real-time PCR analysis we chose genes associated mainly with metabolic processes and ion transport. Significant change in expression was observed for all selected genes (*slc26a4*, *rps7*, *slc5a2*, *aqp6*, *plcd1*, *rny3*, *kcnk5*, *slc6a6*, *atp6v1g3*) (Figure 5). Several genes had increased expression in *insrr* knockout mice under normal condition such as *rny3*, *kcnk5*, *slc6a6*, *atp6v1g3*. Some of them (*rny3*, *kcnk5*, *slc6a6*) had also significantly increased expression after bicarbonate treatment. Reduced expression in *insrr* knockout animals was observed for *slc26a4*, *rps7*, *slc5a2*, *plcd1*, *aqp6*. Of these, *slc26a4*, *slc5a2* decreased their expression in *insrr* knockout mice under bicarbonate loading. (Figure 5). We confirmed that gene *kcnk5* significantly changed expression in kidney of wild-type mice after bicarbonate loading, but we did not find a statistically significant change in expression in the kidneys of knockout animals.

*Slc5a2* gene encodes sodium-glucose cotransporter-2 (SGLT2) which is involved in proximal tubule glucose reabsorption. Inhibitors of the SGLT2 protein are used as mild hypoglycemic agent in diabetes mellitus [25]. In *insrr* knockout animals, *slc5a2* expression decreases by 1.6 times according to NGS data and by 2.7 times according to real-time PCR data. Thus, we tested whether a decrease in *slc5a2* expression affects blood glucose in *insrr* knockout animals. We performed intraperitoneal glucose tolerance test (IPGTT) in *insrr* knockout and wild type animals (seven mice per group). Two hours after glucose administration, we did not observe a significant difference in the concentration of glucose in the blood of wild-type and knockout animals. Surprisingly, at 15, 30, and 60 min, the blood glucose concentration of knockout animals was higher than in wild-type mice. Thus, the mean of the area under the curve (AUC) was $1428 \pm 50.24$ for wild type mice and $1786 \pm 132.0$ for knockout mice with statistical difference $p < 0.05$ (Figure 6).

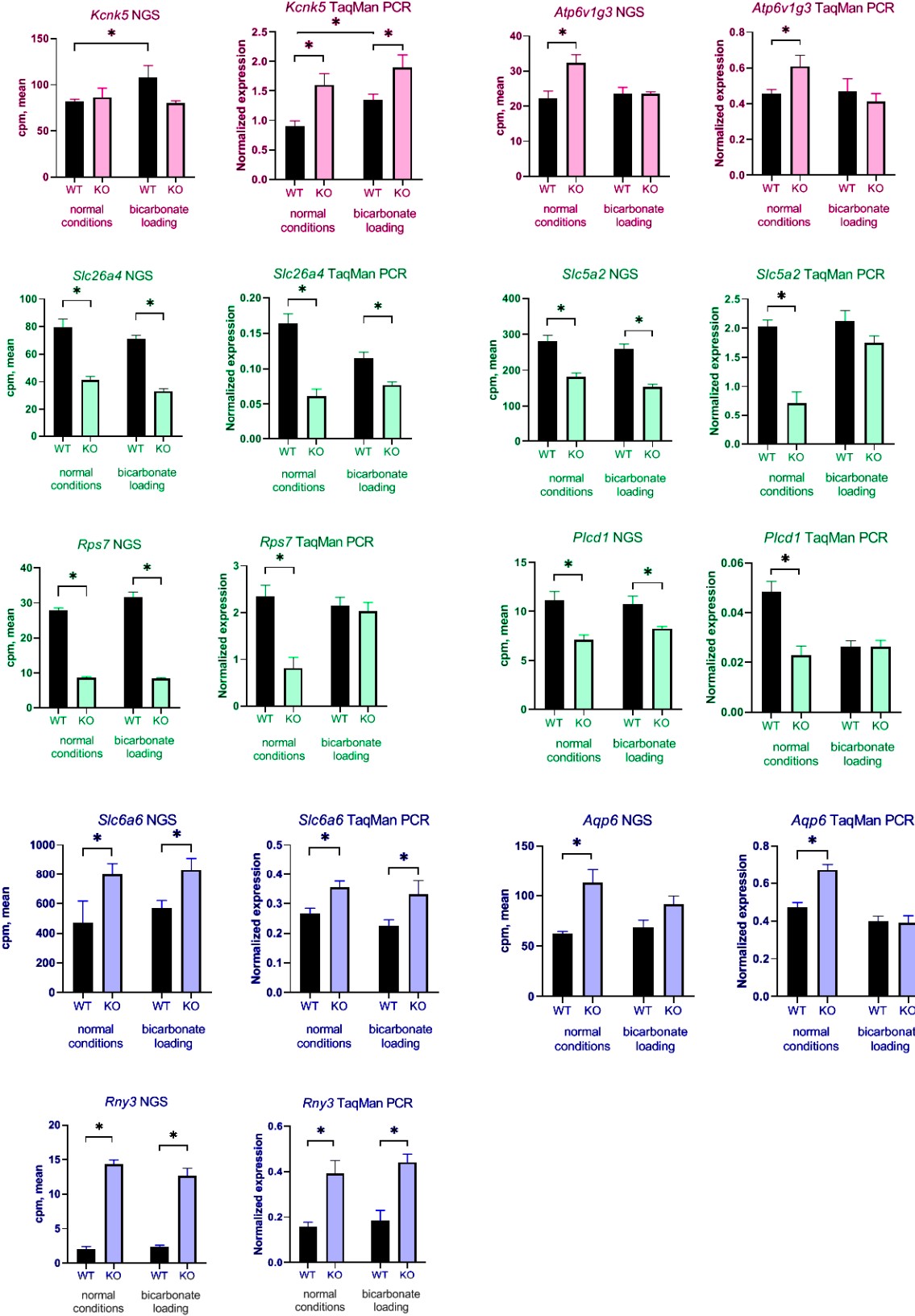

**Figure 5.** Comparison of NGS data and TaqMan qPCR. Pink bars—genes that change their expression after bicarbonate loading (pH-sensing), green bars—genes that decreased their expression after insrr knockout, blue bars—genes that increased their expression after *insrr* knockout. * *p*-value < 0.05 compared to reference gene.

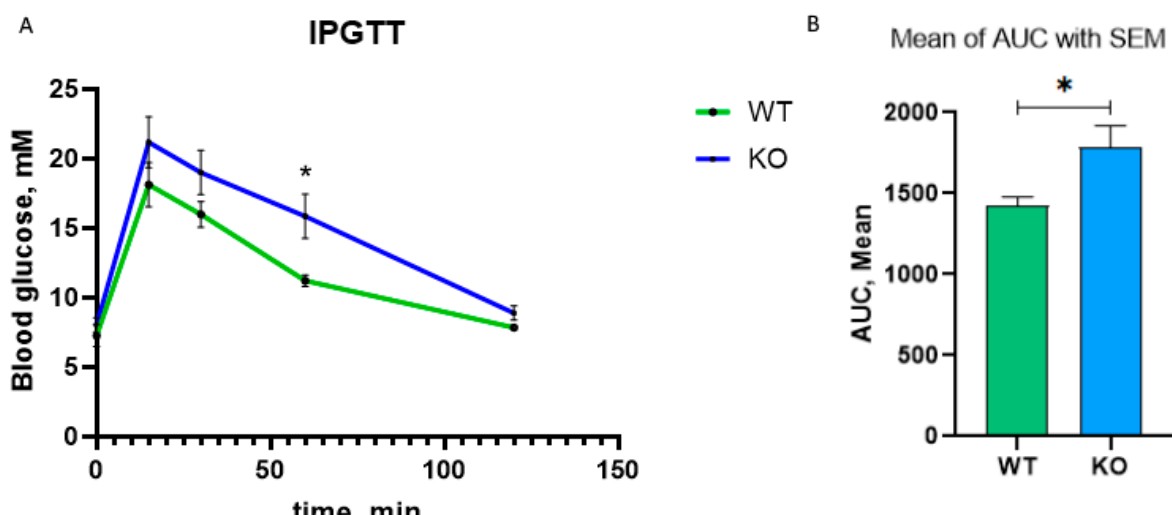

**Figure 6.** Intraperitoneal glucose-tolerance test. (**A**) Curve for blood glucose concentration in different time points. (**B**) Areas under curve for two genotypes, green—wild type, blue—knock-out IRR * $p < 0.05$.

## 4. Discussion

Recently, using morpholino-induced knockdown of ortholog IRR in *Xenopus laevis* embryos with bulk RNA-seq of embryos at the middle neurula stage we demonstrated that *insrr* downregulation leads to development retardation and elicited a general shift of expression towards genes specifically expressed before and at the onset of gastrulation [26]. Here using a similar approach, we try to find and detect genes differentially expressed in the kidney in the absence of IRR, as well as renal genes regulated via bicarbonate loading and depending upon IRR expression. After GO-enrichment, we found, that the absence of IRR leads to changes in the expression of genes associated with metabolic processes. Genes involved in oxidative phosphorylation and ribosomal proteins are substantially upregulated after IRR knockout (such as *ndufa1*, *mt-nd6*, *cox6b*, *atp6v1g3*), whereas several membrane transporter genes were rather downregulated (such as *slc26a4*, *slc16a1*, *slc5a2*). It indicates that IRR signaling is involved in energy supply of cells. While in the knockout group bicarbonate treatment caused no substantial effect according to fgsea; in wild-type mice it resulted in perfectly logical alteration of transporter activity, which is considerably higher in the bicarbonate group. The explanation of such different responses on bicarbonate treatment in knockout and wild-type groups is that *insrr* knockout itself causes changes comparable to those of bicarbonate. Although IRR is well-known expressed in β-intercalated cells of the cortical collecting duct we found changes in gene expression in distant parts of nephron including proximal tubule. Since most of the bicarbonate reabsorption occurs in the proximal tubules, the change in gene expression in PT cells may represent an adaptive response to an increase of bicarbonate concentration in the blood of *insrr* knockout animals. We found changes in expression of *kcnk5*, *slc6a6*, *slc5a2* genes coding membrane transporters and channels. Chen et al. provided a database (Available online: https://esbl.nhlbi.nih.gov/MRECA/Nephron/ (accessed on 4 March 2021)) [27] where we localized the genes whose expression, according to our sequencing data, was reduced. There were several genes *plcd1*, *rps7* and *slc5a2* that decreased their expression after *insrr* knockout, but none of it localized in beta-intercalated cells [28]. For plcd1 the region with the highest occurrence of the transcript is a thick, ascending limb of loop of Henle, for *rps7* it is a descending limb of loop of Henle, and for slc5a2 it is proximal tubule. Thus, for the first time, we showed a change in the pattern of the gene expression not only in β-cells, but also in the entire kidney (Figure 7).

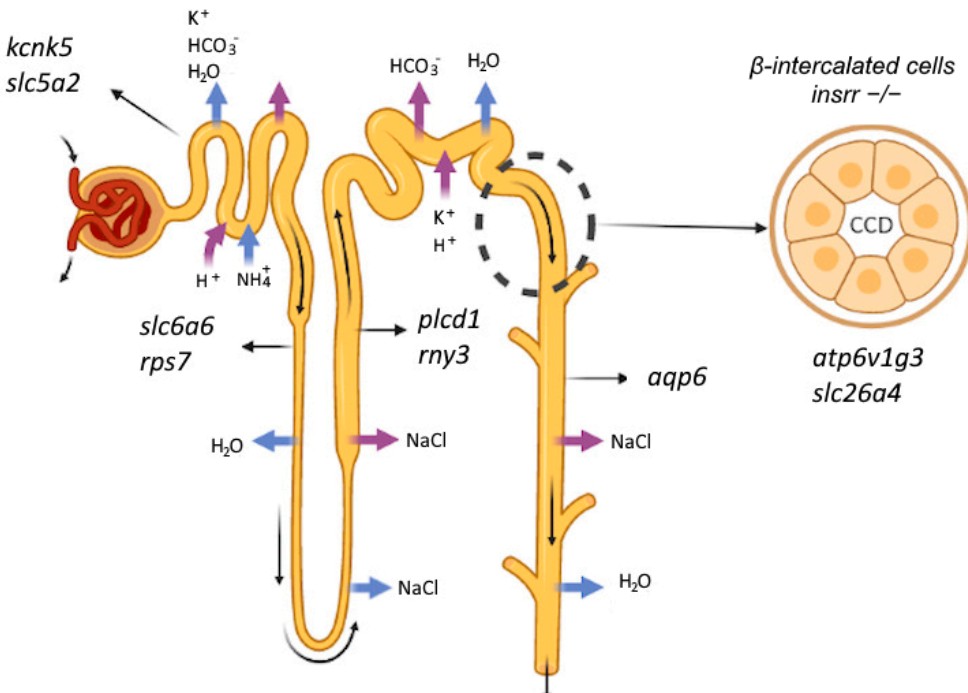

**Figure 7.** Knockout of *insrr* gene and changes in nephron.

Surprisingly, we found that expression of *kcnk5*, which encoding potassium two pore domain channel subfamily K member 5 (TASK2), is significantly increased in wild-type mice kidneys after bicarbonate loading but not in the kidneys of knockout mice. TASK2 is a pH sensing molecule that is activated by an extracellular alkalinization, and it is expressed at the basolateral side of proximal tubule cells. The main functions of TASK2 in kidneys are bicarbonate and water reabsorption, and cell volume regulation. TASK2 deletion in the kidney causes bicarbonate loss in the urine, leading to metabolic acidosis. TASK2 KO mice also exhibit significantly reduced arterial blood pressure compared to WT mice [29]. TASK2 functionally coupled with $Na^+$/ $HCO_3^-$ cotransporter NBCe1-A. The efflux of $Na^+$ and $HCO_3^-$ ions by NBCe1-A leads to depolarization of the basolateral membrane [30]. Increased $HCO_3^-$ concentration in the extracellular space causes the rise in pH that activates TASK2. The deletion of NBCe1-A as well as TASK2 causes metabolic acidosis [29]. Thus, increased expression *kcnk5* may represent adaptive response for elevated concentration of $HCO_3^-$ in both *insrr* knockout mice and wild-type mice under bicarbonate loading.

Also, we found an increased expression of *slc6a6* encoding $Na^+$- and $Cl^-$–dependent taurine transporter (TauT). The main role of taurine transport is osmoregulation which contributes to cell volume regulation [31]. Taurine transporter-deficient mice exhibit impaired energy metabolism, as evidenced by dysfunction of the respiratory chain in various organs [32]. The elevated level of slc6a6 expression is in agreement with the overall upregulation of the genes involved in oxidative phosphorylation in *insrr* knockout mice.

$Na^+$/glucose cotransporter 2 (SGLT2) encoded by *slc5a2* is a main glucose transporter in kidneys. Ninety percent of glucose is reabsorbed in proximal tubule cells via SGLT2 [33]. And according to our dataset the expression of $Na^+$/glucose cotransporter 1 (SGLT1) does not change at all (see Supplement Data File S1). Since *slc5a2* is downregulated, we performed intraperitoneal glucose tolerance test. Surprisingly, we found that at 60 min, the concentration of glucose in the blood of knockout animals was higher than in wild-type mice and AUC was significant higher in knockout mice. Thus, we have shown different regulation of glucose concentration in *insrr* knockout littermates mice compared to wild-type mice. The fact that glucose levels are higher and SGLT2 levels are reduced in knockout mice suggests that the decrease in SGLT2 levels may be a response to the hyperglycemia that occurs in knockout mice.

Among the genes expressed in collecting duct (CD), we detected a change in the expression of the *slc26a4*, *aqp6*, *atp6v1g3*, *slc26a4* found only in β-intercalated cells, aqp6 in α-intercalated cells, *atp6v1g3* in both types of cells in CD.

In general, conducting a study of the transcriptome of *insrr* knockout animals reveals profound differences in fundamental processes in the cell, such as energy metabolism and protein synthesis; the influence of which probably affects the phenotype much more than we think and observe. For the first time in this work, a mouse kidney transcriptome was analyzed under two physiological states, basal state, and bicarbonate load, which made it possible to identify genes associated with both the regulation of acid–base balance and insulin receptor-related receptor function.

**Supplementary Materials:** The following supporting information can be downloaded at: https://www.mdpi.com/article/10.3390/cimb45120606/s1, Figure S1: GO terms for which the IRR knockout differential genes under bicarbonate treatment; Figure S2: GO terms for which the DE genes from the turquoise WGCNA module; Figure S3: GO terms for which the DE genes from the blue WGCNA module; Table S1: Top 25 DEGs which changes expression in mouse kidney after insrr knockout.; Table S2: Top 25 DEGs which IRR dependently change expression in mouse kidney under bicarbonate loading. Data file S1: Comparative analysis of the kidney transcriptomes of wild-type (WT) and *insrr* knockout (KO) mice using RNAseq; Data file S2: Data for Figure 4.

**Author Contributions:** Conceptualization, A.G.P. and I.E.D.; methodology, E.A.G. and O.V.S.; software, D.M.B.; validation, D.M.B.; formal analysis, D.M.B.; investigation, E.A.G. and O.V.S.; resources, D.E. and I.E.D.; data curation, A.V.E. and I.E.D.; writing—original draft preparation, E.A.G. and I.E.D.; writing—review and editing, A.V.E., E.A.G., O.V.S., D.M.B., D.E. and I.E.D.; visualization, E.A.G., D.M.B. and I.E.D.; supervision, A.G.P., A.V.E. and I.E.D.; project administration, A.G.P. and I.E.D.; funding acquisition I.E.D. All authors have read and agreed to the published version of the manuscript.

**Funding:** This work was financially supported by the Russian Science Foundation (grant number. 23-25-00298).

**Institutional Review Board Statement:** All experiments with animals were performed according to the protocol of the Institutional Animal Care and Use Committee (IACUC), approved by the Bioethics Commission of the Shemyakin–Ovchinnikov Institute of Bioorganic Chemistry of the Russian Academy of Sciences (IBCH RAS) (project identification code No. 341), and handled in accordance with the Animals (Scientific Procedures) Act 1986 and Helsinki Declaration.

**Informed Consent Statement:** Not applicable.

**Data Availability Statement:** Raw RNA-seq data were deposit to European Nucleotide Archive under GSE200638 number (Available online: https://www.ncbi.nlm.nih.gov/geo/query/acc.cgi?acc=GSE200638 (accessed on 13 April 2022)).

**Conflicts of Interest:** The authors declare no conflict of interest.

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
