# Peer review of "A Comparative Kidney Transcriptome Analysis of Bicarbonate-Loaded insrr-Null Mice"

_cimb, doi:10.3390/cimb45120606_

Round 1

Reviewer 1 Report

Comments and Suggestions for Authors

The article used transcriptome analysis in insurr-null mice. It is well-written and some minor comments are as follows.

1. Page 1, line 2: insrr should be italic, just like the gene name in line 17.

2. Page 1, line 23: the author can consider to put an ‘and’ in between alc6a6 and atp6v1g3. All gene names should be italic including kcnk5 (line 24).

3. Page 1, line 35: CO2/HCO3-; 2 and 3 should be subscripted and – should be superscripted. Please check all chemical names of whole manuscript.

4. Page 3: Invitrogen (line 103), Agilent (line 104), Ambion (line 105) and NEB (line 106); please add both city and country names.

5. Page 3: STAR (line113), edgeR (line 115) and BioMart (line 123); are they softwares? If yes, please provide company name, city name and country name.

6. Page 4, lines 198-204: in this paragraph, WT and KO are abbreviations in line 198; however, you put the full names in lines 203 and 204. In general, it is better to have the full name at its first appearance.

7. Page 5, line 217: the author used DE genes here, but DEGs in the main text (lines 205-215). I understand that DE genes equals to DEGs. If possible, you should try to use consistent name.

8. Page 5, Figure 1 Legend: the authors should explain what PC1 and PC2 stand for? Also disccuss them in the main text.

9. Page 8, Figure 5: I see ‘*’ in subfigures. Please explain it in the legend.

10. Page 11, line 318: Xenopus laevis should be italic.

11. Page 12, line 350: insrr gene should be italic.

12. References need to be carefully edit. For example, in line 437, is this Russia letters necessary or not?

Author Response

Dear Reviewer,

We would like to express our gratitude for taking the time to review our article on transcriptome analysis in insrr-null mice. Your insightful feedback has been invaluable, and we are committed to thoroughly addressing each of the points you raised in the revised manuscript.

Here are the specific enhancements we have made in response to your feedback:

  1. Italized "insrr" in line 2 on page 1.
  2. Added "and" between "alc6a6" and "atp6v1g3" and ensured appropriate italicization of all gene names, including "kcnk5" on line 24 of page 1.
  3. Carefully reviewed the subscripting and superscripting of chemical names throughout the manuscript to ensure accuracy.
  4. Added city and country names for Invitrogen, Agilent, Ambion, and NEB in lines 103-106 on page 3.
  5. STAR is an academic software, for which we cite an article in which it was introduced. This software is cited in the same way by its author, Prof. Alexander Dobin, in his articles (see, for instance, org/10.1101/gr.275613.121). edgeR is an R package, which we cite as indicated by its authors in the package itself (see citation("edgeR")). For BioMart, we have updated our citation according to the Ensembl official recommendations (http://www.ensembl.org/info/about/publications.html).
  6. Maintained consistent use of abbreviations for "WT" and "KO" throughout the paragraph on lines 198-204 of page 4.
  7. Ensured the consistent use of terminology, "DE genes" or "DEGs," throughout the manuscript on line 217 of page 5.
  8. PC1 and PC2 in this context represent Principal Component 1 and Principal Component 2, which are commonly used abbreviations in data analysis. Principal components are orthogonal axes that capture the most significant variation in the data. In Figure 1B, they are used to visualize the distribution of the samples. The absolute values and signs of these components are not inherently meaningful for the analysis. What is important is the relative position and linear (non)-separability of different sample groups on the plane given by the first two principal components.
  9. Included an explanation for the asterisk (*) seen in the subfigures of Figure 5 in the legend on page 8.
  10. Italized "Xenopus laevis" as per the correct formatting on line 318 of page 11.
  11. Italized "insrr" in line 350, maintaining consistency throughout the manuscript on page 12.
  12. Carefully edited and formatted the references, ensuring accuracy and correctness, including the necessity of non-English letters.

We appreciate the time you took to provide us with such valuable feedback, and we are looking forward to submitting the revised manuscript addressing these comments.

Reviewer 2 Report

Comments and Suggestions for Authors

General comments:

The purpose of the study was to examine the effects of bicarbonate loading on gene expressions in normal mice versus insrr knockout mice. Insrr gene encodes the receptor tyrosine kinase IRR (insulin receptor-related receptor), which is largely expressed in renal β -intercalated cells and involved in alkali sensing due to its ability to autophosphorylate under alkalization (pH > 7.9). In this study, the authors compared the renal transcriptomes of wild type mice and insrr knockout mice under basal or bicarbonate-loaded conditions and found differential regulation of a number of genes in the kidneys. They also found that insrr knockout affects the expression of genes related mostly to the ATP metabolic and electron transport chain processes. The study is interesting and provides valuable information on the expressional changes in renal transcriptomes associated with IRR. There are weaknesses. Lack of clarity in writing makes it a bit hard to read the manuscript. The purpose of the study was described and the results were obtain, but the interpretation and discussion regarding the functional and physiological implication of bicarbonate loading for the gene expression changes are limited. Some experiments do not appear to be designed well and made the conclusion unclear. Specific comments are written below.

Specific comments:

Line 19. Rewrite the sentence ‘Which other kidney mechanisms are under the control of IRR…’.

Line 42-44. ‘All the filtered HCO3 is reabsorbed by proximal tubules cells.’ This is not true. About 70% of filtered HCO3 is reabsorbed in the proximal tubules, 25% in the thick ascending limb, and the remaining 5% in the collecting duct. HCO3 is also produced in the kidney when blood pH is low.

Line 55. what is IRS-1?

Line 88. Explain why female only was used in this study.

Line 94. Mice were fasted for 12 h before kidney collection. Fasting affects many genes. The authors should explain how the transcriptome data in this study were affected by fasting. Add the explanation before line 205.

Line 189-197.  This paragraph should be moved to the Materials and Methods.

Line 202-203. Change ‘physiological’ to ‘normal’ and ‘alkali-loading’ to ‘alkali-loaded’.   

Missing data: It is unclear whether bicarbonate loading induced IRR autophosphorylation that occurs under alkalization (pH 7.9). Include data of blood pH, bicarbonate, and other electrolyte values in four groups of mice. Also, include autophosphorylation data.

Line 217. Figure 1B needs clarification. What do PC1 and PC2 indicate and what are the meanings of negative and positive values?

Line 289 and Figure 5. Include the reference ap3d1 in NGS and TaqMan PCR.

Line 305-316. Include the glucose tolerance data (Figure S4) as a new main figure. Alternatively, this can be deleted because it does not provide much information on addressing the reason of Slc5a2 downregulation in knockout mice.   

Line 379. The increased plasma glucose levels in insrr knockout mice are opposite to previous reports that plasma glucose levels in mice are reduced by SGLT2 inhibition and in Sglt2 knockout mice. The authors need to discuss this inconsistency. Elaborate ‘a compensatory response to elevated blood glucose levels in insrr knockout mice’.

Line 385-398. This paragraph is unrelated to the gene expressional changes in insrr knockout mice. Rewrite or delete.

Comments on the Quality of English Language

Moderate editing of English language required.

Author Response

Dear Reviewer,

We would like to express our gratitude for taking the time to review our article on transcriptome analysis in insrr-null mice. Your insightful feedback has been invaluable, and we are committed to thoroughly addressing each of the points you raised in the revised manuscript.

Here are the specific enhancements we have made in response to your feedback:

Specific comments:

Line 19. Rewrite the sentence ‘Which other kidney mechanisms are under the control of IRR…’

The sentence has been revised to “The specific kidney mechanisms that are under the control of IRR remain unknown.”

Line 42-44. ‘All the filtered HCO3 is reabsorbed by proximal tubules cells.’ This is not true. About 70% of filtered HCO3 is reabsorbed in the proximal tubules, 25% in the thick ascending limb, and the remaining 5% in the collecting duct. HCO3 is also produced in the kidney when blood pH is low.

Thank you for your input. The statement regarding the reabsorption of filtered HCO3 has been accurately revised in line 42-44. It now reflects that around 70% of filtered HCO3 is reabsorbed in the proximal tubules, 25% in the thick ascending limb, and the remaining 5% in the collecting duct.

Line 55. what is IRS-1?

 It is insulin receptor substrate 1, that had explained in text, thank you.

Line 88. Explain why female only was used in this study.

 In this work, we used female mice because they are more convenient to work with than males. Since males behave very aggressively, which makes it very difficult to work with them.

Line 94. Mice were fasted for 12 h before kidney collection. Fasting affects many genes. The authors should explain how the transcriptome data in this study were affected by fasting. Add the explanation before line 205.

Eating has dramatic effects on acid-base balance an its regulation. Eating is in general an acid load, however the post prandial period is also characterized by a marked alkalosis known as alkali tide. Therefore, to limit the cofounding parameters linked to eating experiments that aim to characterizes chronic renal regulation generally are performed after an overnight fasting.

We added this text to our article:

Also, in order to avoid additional acid load due to the feeding process of mice, which can lead to additional variation in the acid-base status of mice, mice were fasted for 12 hours with free access to water (or NaHCO3 solution).

Line 189-197.  This paragraph should be moved to the Materials and Methods.

 The paragraph has been moved in “Material and Methods”

Line 202-203. Change ‘physiological’ to ‘normal’ and ‘alkali-loading’ to ‘alkali-loaded’.   

 We have made these corrections to the text.

Missing data: It is unclear whether bicarbonate loading induced IRR autophosphorylation that occurs under alkalization (pH 7.9). Include data of blood pH, bicarbonate, and other electrolyte values in four groups of mice. Also, include autophosphorylation data.

Thank you for your valuable comment. However, we have already published these data for knockout and wild-type mice in these articles (Deyev, I.E.; Sohet, F.; Vassilenko, K.P.; Serova, O. V.; Popova, N. V.; Zozulya, S.A.; Burova, E.B.; Houillier, P.; Rzhevsky, D.I.; Berchatova, A.A.; et al. Insulin Receptor-Related Receptor as an Extracellular Alkali Sensor. Cell Metab 2011, 13, 679–689     and  Deyev IE; Rzhevsky DI; Berchatova AA; Serova OV; Popova NV; Murashev AN; Petrenko AG Deficient Response to Experi-mentally Induced Alkalosis in Mice with the Inactivated Insrr Gene. Acta Naturae 2011, 3, 2011). Our article (Cell Metabolism, 2011) also shows IRR autophosphorylation by bicarbonate-loading condition in kidney

Line 217. Figure 1B needs clarification. What do PC1 and PC2 indicate and what are the meanings of negative and positive values?

 PC1 and PC2 in this context represent Principal Component 1 and Principal Component 2, which are commonly used abbreviations in data analysis. Principal components are orthogonal axes that capture the most significant variation in the data. In Figure 1B, they are used to visualize the distribution of the samples. The absolute values and signs of these components are not inherently meaningful for the analysis. What is important is the relative position and linear (non)-separability of different sample groups on the plane given by the first two principal components.

Line 289 and Figure 5. Include the reference ap3d1 in NGS and TaqMan PCR.

Thanks for the note, however, presenting TaqMan PCR histograms for the reference gene does not seem rational from the point of view of the authors of the article, since the values are normalized to this gene.

name

KO1_H2O

KO2_H2O

KO3_H2O

KO4_H2O

KO5_NaHCO3

KO6_NaHCO3

KO7_NaHCO3

KO8_NaHCO3

Ap3d1

1705,3

1573,9

1590,8

1719,8

1765,7

1702,3

1804,0

1712,3

name

WT1_H2O

WT2_H2O

WT3_H2O

WT4_H2O

WT5_NaHCO3

WT6_NaHCO3

WT7_NaHCO3

WT8_NaHCO3

Ap3d1

1395,6

1355,9

1396,1

1455,5

1439,3

1516,9

1575,1

1463,0

The level of gene expression (cpm) by NGS in mice groups:

Also, below the example of qPCR raw data (Ct) of reference gene Ap3d1

Ct mean

Ct mean

Ct mean

Ap3d1_WT_water

19,26667

18,4

18,53333

Ap3d1_WT_bicarb

19,63333

19,06667

18,6

Ap3d1_KO_water

21,6

18,56667

18,53333

Ap3d1_KO_bicarb

20,83333

18,06667

20,36667

Line 305-316. Include the glucose tolerance data (Figure S4) as a new main figure. Alternatively, this can be deleted because it does not provide much information on addressing the reason of Slc5a2 downregulation in knockout mice.   

 Figure S4 has been moved in a new main figure.

Line 379. The increased plasma glucose levels in insrr knockout mice are opposite to previous reports that plasma glucose levels in mice are reduced by SGLT2 inhibition and in Sglt2 knockout mice. The authors need to discuss this inconsistency. Elaborate ‘a compensatory response to elevated blood glucose levels in insrr knockout mice’.

Many thanks for this comment. We rewrite part of discussion about expression SGLT2 and changed and changed our explanation to this “The fact that glucose levels are higher and SGLT2 levels are reduced in knockout mice suggests that the decrease in SGLT2 levels may be a response to the hyperglycemia that occurs in knockout mice”

Line 385-398. This paragraph is unrelated to the gene expressional changes in insrr knockout mice. Rewrite or delete.

Thanks for your comment. We have removed this part of the discussion.

Reviewer 3 Report

Comments and Suggestions for Authors

A constant and major problem in critical ill patients that often associate multiple-organ deficiency is to maintain normal values of plasma pH and bicarbonate concentration. Considering that the kidneys play an important role in acid-base regulation, especially through IRR expression, the present study highlights the biological processes that may be affected by IRR in the kidneys and could offer new insights of this issue. In addition, based on the results, a mapping of the gene expression (involved in this regulation) in the entire kidney has been provided. The methodology and results were clearly described, and the conclusions were supported by the study findings. I congratulate you for the quality of the presentation and, especially, for the new data provided by your research, which can improve our current knowledge.

Author Response

We would like to express our gratitude for taking the time to review our article on transcriptome analysis in insrr-null mice. 

Reviewer 4 Report

Comments and Suggestions for Authors

1. Select one or two kidney-related cell lines for studying the specific gene(s) of interest. Use techniques like siRNA or CRISPR/Cas9 to knock down the gene(s) and observe changes in intracellular acid-base balance and energy metabolism-related parameters. This approach will help elucidate the gene's function in specific cell types.

2. Utilize techniques such as immunohistochemistry or in situ hybridization to examine the expression and distribution of the key gene in the kidney tissues of the knockout mice. This will help determine the gene's expression patterns in different renal cell types, providing valuable insights into its role in kidney function.

3. If applicable, perform functional rescue experiments by reintroducing the knocked-out gene(s) into the knockout cells or mice. Assess whether restoring the gene expression reverses the observed phenotypic changes, confirming the direct impact of the gene(s) on acid-base balance and energy metabolism.

4. Subject knockout mice to specific physiological challenges, such as acidosis or high-fat diet feeding, and monitor their responses. Compare these responses to those of wild-type mice under similar conditions. Analyze parameters related to acid-base balance, metabolic rate, and substrate utilization to investigate the gene's role in adapting to physiological stressors.

5. If experimental methods are described, ensure each step is thoroughly explained, allowing other researchers to replicate your experiments. Providing sufficient detail enhances the replicability of your study.

6. In the Discussion section, deeply analyze your research results, comparing them with previous studies. Address the limitations of your study and propose suggestions for future research. This in-depth analysis enhances the scholarly value of your article.

Comments on the Quality of English Language

 Moderate editing of English language required

Author Response

  1. Select one or two kidney-related cell lines for studying the specific gene(s) of interest. Use techniques like siRNA or CRISPR/Cas9 to knock down the gene(s) and observe changes in intracellular acid-base balance and energy metabolism-related parameters. This approach will help elucidate the gene's function in specific cell types.

Thank you very much for this comment. Indeed, to study it in more detail, we need to make knockout cell lines from kidneys. We plan in the near future to make various knockout cell lines with a number of genes that were changed in our knockout mice.

  1. Utilize techniques such as immunohistochemistry or in situ hybridization to examine the expression and distribution of the key gene in the kidney tissues of the knockout mice. This will help determine the gene's expression patterns in different renal cell types, providing valuable insights into its role in kidney function.

We have also additionally planned in the near future experiments to determine changes not only in the level of expression, but also possibly changes in the localization of various transporters (like pendrin) using immunohistochemistry in the kidneys of mice under different conditions.

  1. If applicable, perform functional rescue experiments by reintroducing the knocked-out gene(s) into the knockout cells or mice. Assess whether restoring the gene expression reverses the observed phenotypic changes, confirming the direct impact of the gene(s) on acid-base balance and energy metabolism.

Thank you for this note. By really using reintroducing the knocked-out gene, we will be able to better study and understand the mechanisms of insrr gene  action in the regulation of acid-base balance.

  1. Subject knockout mice to specific physiological challenges, such as acidosis or high-fat diet feeding, and monitor their responses. Compare these responses to those of wild-type mice under similar conditions. Analyze parameters related to acid-base balance, metabolic rate, and substrate utilization to investigate the gene's role in adapting to physiological stressors.

Also, over the next year, we wanted to obtain new results on the effect of insrr gene knockout when feeding mice by high-fat diet.

  1. If experimental methods are described, ensure each step is thoroughly explained, allowing other researchers to replicate your experiments. Providing sufficient detail enhances the replicability of your study.

We certainly agree with you and we describe our experimental methods and approaches in detail in our article.

  1. In the Discussion section, deeply analyze your research results, comparing them with previous studies. Address the limitations of your study and propose suggestions for future research. This in-depth analysis enhances the scholarly value of your article.

Of course, the results of our work should be fully and thoroughly discussed in our article. We did our best.

Round 2

Reviewer 2 Report

Comments and Suggestions for Authors

1. The authors' rationale to use female mice only is a lame excuse and lacks scientific justification. It is also somewhat disrespectful for researchers who use male mice only. 

2.  The authors stated in line 298-299 that ap3d1 expression did not change in animals of two genotypes under different conditions according to NGS data. However, the ap3d1 gene expression level (cpm), provided by the authors in response to my request, shows a significant difference (p < 0.01) between WT and KO mice. Thus, the statement in line 298-299 is false. This raises a concern of the manuscript integrity.     

Comments on the Quality of English Language

n/a

Author Response

Dear Reviewer,

We appreciate your very valuable comments. Our responses to these comments are provided in the attached file.

Reviewer 4 Report

Comments and Suggestions for Authors

No comments

Comments on the Quality of English Language

Minor editing of English language required

Author Response

Thank you for this important note. We have made additional minor editing to the text of our manuscript.

Round 3

Reviewer 2 Report

Comments and Suggestions for Authors

n/a